# Ecological determinants of Cope's rule and its inverse

Shovonlal Roy [1✉], Åke Brännström[2,3,4] & Ulf Dieckmann [2,4,5]

Cope's rule posits that evolution gradually increases the body size in lineages. Over the last decades, two schools of thought have fueled a debate on the applicability of Cope's rule by reporting empirical evidence, respectively, for and against Cope's rule. The apparent contradictions thus documented highlight the need for a comprehensive process-based synthesis through which both positions of this debate can be understood and reconciled. Here, we use a process-based community-evolution model to investigate the eco-evolutionary emergence of Cope's rule. We report three characteristic macroevolutionary patterns, of which only two are consistent with Cope's rule. First, we find that Cope's rule applies when species interactions solely depend on relative differences in body size and the risk of lineage extinction is low. Second, in environments with higher risk of lineage extinction, the recurrent evolutionary elimination of top predators induces cyclic evolution toward larger body sizes, according to a macroevolutionary pattern we call the recurrent Cope's rule. Third, when interactions between species are determined not only by their body sizes but also by their ecological niches, the recurrent Cope's rule may get inverted, leading to cyclic evolution toward smaller body sizes. This recurrent inverse Cope's rule is characterized by highly dynamic community evolution, involving the diversification of species with large body sizes and the extinction of species with small body sizes. To our knowledge, these results provide the first theoretical foundation for reconciling the contrasting empirical evidence reported on body-size evolution.

[1] Department of Geography and Environmental Science, University of Reading, Whiteknights, Reading RG6 6DW, UK. [2] Advancing Systems Analysis Program, International Institute for Applied Systems Analysis (IIASA), Schlossplatz 1, A-2361 Laxenburg, Austria. [3] Department of Mathematics and Mathematical Statistics, Umeå University, 90187 Umeå, Sweden. [4] Complexity Science and Evolution Unit, Okinawa Institute of Science and Technology Graduate University (OIST), 1919-1 Tancha, Onna, Kunigami, Okinawa 904-0495, Japan. [5] Department of Evolutionary Studies of Biosystems, The Graduate University for Advanced Studies (Sokendai), Hayama, Kanagawa 240-0193, Japan. ✉email: shovonlal.roy@reading.ac.uk

Cope's rule states that lineages evolve toward larger body sizes over evolutionary time[1,2]. Recognizing that larger body sizes often improve an animal's ability to capture prey, avoid predators, fight competitors, maintain metabolism, raise thermal inertia, accommodate climatic variation, withstand starvation, extend longevity, attract mates, and enhance reproductive success[3,4], researchers have hypothesized that Cope's rule applies to all animals—and to all mammals in particular[1,5–7]. However, empirical evidence concerning body-size evolution is remarkably conflicting. While many studies of the fossil record have supported Cope's rule[8–18], many studies of extant species have failed to demonstrate systematic increases in body size[19–22]. Specifically, Cope's rule has been reported to apply in species as diverse as North American fossil mammals[9], dinosaurs[11], Paleozoic brachiopods[15], mesozoic birds[13], and marine mammals[17]. In contrast, Cope's rule has been reported not to apply in other species, such as island lizards[23], Alaskan horses[24], certain freshwater fish[19], cryptodiran turtles[20], certain extant mammals[21], several vertebrates[22], and insects[25]. These disparate conclusions have fueled a debate about the validity of Cope's rule[26]. As Cope's rule does not hold in all taxa[27], it has been suggested that Cope's rule could be an artifact of selective perception, resulting from singling out lineages that do increase in body size[26,28,29]. Even within taxa for which Cope's rule has been reported to hold, species in certain clades have still been found to have shrunk in body size over evolutionary time[22–24,30]. The mechanisms underlying Cope's rule are also hotly debated[4,10,26,31–36]. Moreover, it has been argued that reported confirmations of Cope's rule may be the result of passive processes in which an initially small ancestor gradually fills out available niche space[2]. Furthermore, early studies of Cope's rule examining the fossil record have been called into question by demonstrating that the standard interpretation of Cope's rule is not invariant under the choice of body-size measure[37]. Cope's rule is occasionally referred to as Depéret's rule since the latter author's formulation, while having been published two decades later, is recognized as clearer and more explicit[38–40].

The apparently contradictory empirical evidence for and against Cope's rule points to a need for resolving the different positions at a higher level. This requires developing a better understanding of the ecophysiological causes of selection pressures on body size[10], the correlation between body size and rates of evolutionary diversification, and the ecological conditions under which these jointly give rise to phylogenetic patterns consistent with Cope's rule. Since the 1990s, several community-evolution models have been developed that combine complex ecological and evolutionary dynamics through niche-based (e.g., ref. [41]), assembly-based (e.g., refs. [42,43]), or evolutionary approaches (e.g., refs. [44–46]). These models can be used to address questions about the causes of selection pressures on body size, resultant evolutionary diversification rates, and the ecological determinants of emerging phylogenetic patterns. To the best of our knowledge, however, no study has analyzed community-evolution models to elucidate Cope's rule and the conditions under which it applies.

Here we investigate and classify macroevolutionary patterns emerging under different ecological and physiological conditions by developing a new process-based community-evolution model, integrating two established modeling approaches that, respectively, account for the ecophysiological implications of body size[44,46–48] and the role of ecological niches[41]. This novel approach allows us to determine when Cope's rule is expected to apply and to identify two distinct macroevolutionary patterns that are both consistent with Cope's rule. Surprisingly, we also find a third characteristic macroevolutionary pattern that has so far not been examined in the paleontological literature. Our findings fill a lacuna in understanding the applicability of Cope's rule in general ecological communities.

To investigate the eco-evolutionary dynamics and determine phylogenies, we develop a deterministic process-based community-evolution model that describes adaptive changes in two quantitative traits, body size and ecological niche. Previous work has shown that adaptations in body size alone enables the emergence of trophically structured communities[44,46] and that adaptations in ecological niche alone enables the emergence of niche partitioning in communities[49–51]. Both of these traits have thus been included, separately from each other, as key functional traits in previous models. It is only recently, however, that these traits have jointly been incorporated into community-evolution models[52].

By examining model communities structured by body size and ecological niche, we can naturally account for all three fundamental mechanisms of trophic and nontrophic competitive interactions: exploitative competition (through which predating species compete indirectly by consuming the same preyed species), apparent competition (through which preyed species compete indirectly by being consumed by the same predating species), and interference competition (through which species compete directly). In our model, body size and ecological niche affect the gains and losses from trophic interactions through predation and the losses from nontrophic interactions through interference competition. Increasing from zero, the difference in the body sizes of two species has two effects: it raises the predation rate of the larger species upon the smaller one (up to an optimal relative body-size difference, after which the predation rate decreases), and it reduces the interference rate between the two species. Increasing from zero the difference in the ecological niches of two species reduces both the predation rate and interference rate. In summary, the predation rate is maximized when the two species have identical niche traits and an optimal relative difference in body size (with the predating species having a much larger body size than the preyed species), while the interference rate is maximized when the two species have identical niche traits and body sizes.

We consider a species to be extant if its population density exceeds a given threshold. Conversely, if a species' population density falls below this threshold, it is considered extinct and is removed from the community. This threshold can thus be interpreted as a measure of extinction risk resulting from sensitivity to demographic and environmental stochasticity.

The two adaptive traits simultaneously evolve under directional selection in all extant species until a local fitness minimum is reached in any one of them, at which point the ancestral population splits into two under the force of negatively frequency-dependent disruptive selection and, hence, evolutionary diversification occurs. The processes of directional evolution, evolutionary diversification, and extinction enable the emergence of complex ecological communities with potentially intricate phylogenies. On this basis, we report how these phylogenies vary with the strength of interference competition and extinction risk, which allows us to identify three characteristic macroevolutionary patterns, as described in the next section.

A full description of our model—including all equations, parameter values, and biological interpretations—is provided in "Methods" below.

## Results

**Evolution in body size alone: Cope's rule**. We first restrict species evolution to changes in body size, assuming—unrealistically—that species otherwise occupy the same ecological niche. Figure 1 and Supplementary Movie S1 show how the evolution then resulting in our model leads to a community with few species. Starting from a single ancestor species with small body

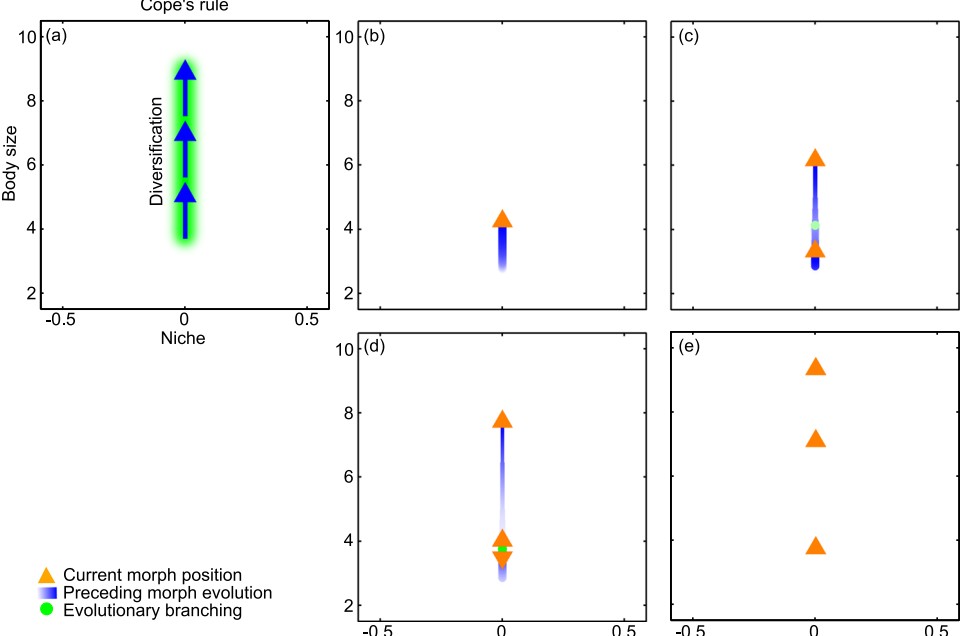

**Fig. 1 Evolution in body size alone: Cope's rule. a** Schematic illustration of phylogenetic pattern. **b–e** Trait combinations in the coevolving community at four successive times. Orange arrowheads show species' current trait values and the current directions of the selection pressures governing their evolutionary change. Blue motion trails represent species' evolutionary trajectories, with their thickness being proportional to species' population densities and with the darkest blue corresponding to the current time. Green circles indicate past trait combinations at which evolutionary diversifications occurred. In this macroevolutionary pattern, evolutionary diversification occurs only in body size: all species then evolve toward larger body sizes until a stable food web of trophic interactions among them emerges. For a full dynamical rendering of the shown community coevolution, see Supplementary Movie S1. Parameters as shown in Table 1.

size, body size initially increases (Fig. 1b), followed by an evolutionary diversification into two species with different body sizes (Fig. 1c). After a slight reduction in body size of the smallest species, a second evolutionary diversification takes place (Fig. 1d), eventually resulting in three species forming an evolutionarily stable community (Fig. 1e). The same qualitative pattern of evolutionary diversification and evolution toward larger body size as illustrated in Fig. 1a also arises for many other parameterizations of our model, with the strength of interference competition showing a strong positive correlation with the number of species in the evolved community[44]: in other words, evolved communities comprise the more species the stronger the interference competition among them. Our model thus shows that when species share a niche, and their ecological interactions are determined by their body size, they primarily evolve toward larger body size, in accordance with Cope's rule.

**Evolution in body size and niche at weak interference competition and low extinction risk: Cope's rule maintained.** We next consider joint evolution in body size and ecological niche. Figure 2 and Supplementary Movie S2 show how, at low strengths of interference competition, a richer evolutionary process than previously described unfolds. Starting again from a single ancestor species, gradual evolution leads to diversification in body size (Fig. 2b) followed by diversification in ecological niche (Fig. 2c). Next, one species evolves to large body size, while the two others evolve to occupy different ecological niches (Fig. 2c). Each species in this latter pair then diversifies in body size (Fig. 2d), and gradual evolution eventually results in an evolutionarily stable community with five species (Fig. 2e). With only a few transient exceptions, evolution in body size is consistent with Cope's rule. The qualitative pattern illustrated in Fig. 2a holds also for many other parameterizations of our model, as long as interference competition is weak and extinction risk is low.

Below, we explore the effects of stronger interference competition after considering the effects of higher extinction risk.

**Evolution in body size and niche with precarious top predators: recurrent Cope's rule.** Several earlier studies have highlighted the positive relationship between a species' extinction risk and body size as an important driver of phylogenetic patterns[14,29,47,53–56]. To capture this relationship, we consider a species to go extinct once its population density falls below a threshold ("Methods"): since populations of larger species have fewer individuals and hence lower population densities, they are more vulnerable to extinction. Figure 3 and Supplementary Movie S3 show how sufficiently high extinction risks qualitatively alter the emergent phylogenetic pattern. Species evolution is initially identical to Fig. 2. As the top predator evolves toward larger body size (Fig. 3b, c), it goes extinct (Fig. 3c). The two next-largest species then evolve to occupy the former top predator's niche, while the two species with lowest trophic levels diversify (Fig. 3d). Next, a further extinction occurs, as only one of the two largest species can succeed in becoming the new top predator. The surviving species continues to evolve toward larger body size, eventually also becoming extinct (Fig. 3e), which perpetuates the cyclic eco-evolutionary dynamics (Fig. 3c–e). Highlighting the cyclic nature of the emergent phylogenetic pattern (Fig. 3a), we refer it as following the recurrent Cope's rule.

**Evolution in body size and niche at strong interference competition: recurrent inverse Cope's rule.** When the strength of interference competition is high (within the range shown in Table 1), a qualitatively different, unexpected phylogenetic history unfolds, as shown in Fig. 4 and Supplementary Movie S4. A species-rich community first emerges from a single ancestor species, through a process akin to that shown in Fig. 3. The

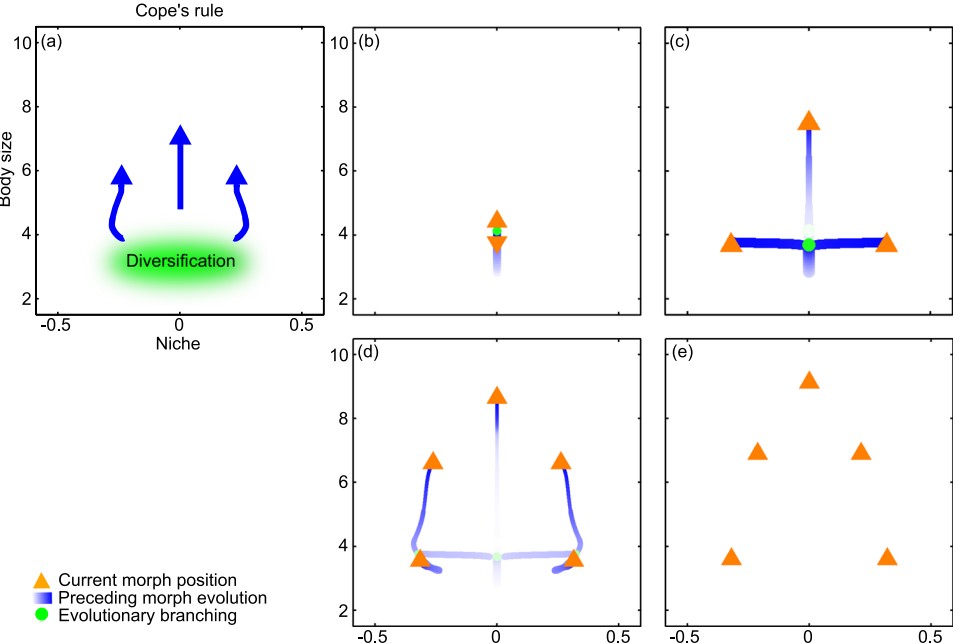

**Fig. 2 Evolution in body size and niche at weak interference competition and low extinction risk: Cope's rule maintained. a** Schematic illustration of the phylogenetic pattern. **b–e** Trait combinations in the coevolving community at four successive times. Orange arrowheads show species' current trait values and the current directions of the selection pressures governing their evolutionary change. Blue motion trails represent species' evolutionary trajectories, with their thickness being proportional to species' population densities and with the darkest blue corresponding to the current time. Green circles indicate past trait combinations at which evolutionary diversifications occurred. In this macroevolutionary pattern, evolutionary diversification occurs in both traits: all species then evolve toward larger body sizes until a stable food web of trophic interactions among them emerges. For a full dynamical rendering of the shown community coevolution, see Supplementary Movie S2. Parameters as shown in Table 1, with $\alpha = 0.002$.

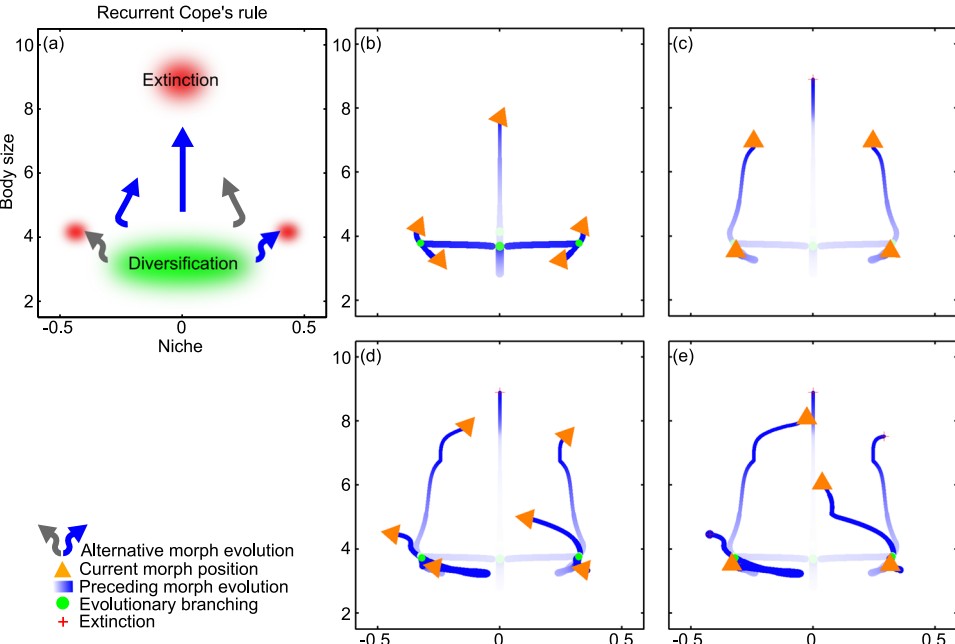

**Fig. 3 Evolution in body size and niche with precarious top predators: recurrent Cope's rule. a** Schematic illustration of phylogenetic pattern. **b–e** Trait combinations in the coevolving community at four successive times. Orange arrowheads show species' current trait values and the current directions of the selection pressures governing their evolutionary change. Blue motion trails represent species' evolutionary trajectories, with their thickness being proportional to species' population densities and with the darkest blue corresponding to the current time. Green circles and red crosses, respectively, indicate past trait combinations at which evolutionary diversifications and species extinctions occurred. In this macroevolutionary pattern, evolutionary diversification occurs at small body sizes: all species then evolve toward larger body sizes, punctuated by the recurrent extinction of top predators. For a dynamic rendering of the shown community coevolution, see Supplementary Movie S3. For a full dynamical rendering of the shown community coevolution, see Supplementary Movie S3. Parameters as shown in Table 1, with $\alpha = 0.002$ and $\epsilon = 0.005$.

**Table 1 Model parameters and their default values.**

| Description | Symbol | Default value |
|---|---|---|
| Maximum rate of predation | $\beta$ | 2.65 |
| Maximum rate of interference competition | $\alpha$ | 0 to 1 |
| Trophic efficiency | $\lambda$ | 0.1 |
| Size of basal autotrophic resource | $s_0$ | 1 |
| Intrinsic growth rate of basal autotrophic resource | $g_0$ | 10 |
| Carrying capacity of basal autotrophic resource | $K_0$ | 100 |
| Extinction threshold | $\epsilon$ | 0–0.01 |
| Maximum rate of intrinsic mortality | $d_0$ | 0.1 |
| Allometric exponent of intrinsic mortality | $q$ | 0.25 |
| Natural logarithm of optimal body-size ratio of predation | $\mu$ | 3 |
| Quadratic dispersion of predation in body size | $\sigma_p$ | 0.75 |
| Quartic dispersion of predation in body size | $\gamma_p$ | 0 |
| Quadratic dispersion of interference competition in body size | $\sigma_c$ | 0.33 |
| Quartic dispersion of interference competition in body size | $\gamma_c$ | 0.457 |
| Quadratic dispersion of predation in ecological niche | $\sigma_P$ | 1.38 |
| Quartic dispersion of predation in ecological niche | $\gamma_P$ | 1.54 |
| Quadratic dispersion of interference competition in ecological niche | $\sigma_C$ | 0.402 |
| Quartic dispersion of interference competition in ecological niche | $\gamma_C$ | 0.18 |
| Rate coefficient of evolutionary change in body size | $\varepsilon_s$ | 0.000025 |
| Rate coefficient of evolutionary change in ecological niche | $\varepsilon_n$ | 0.000025 |

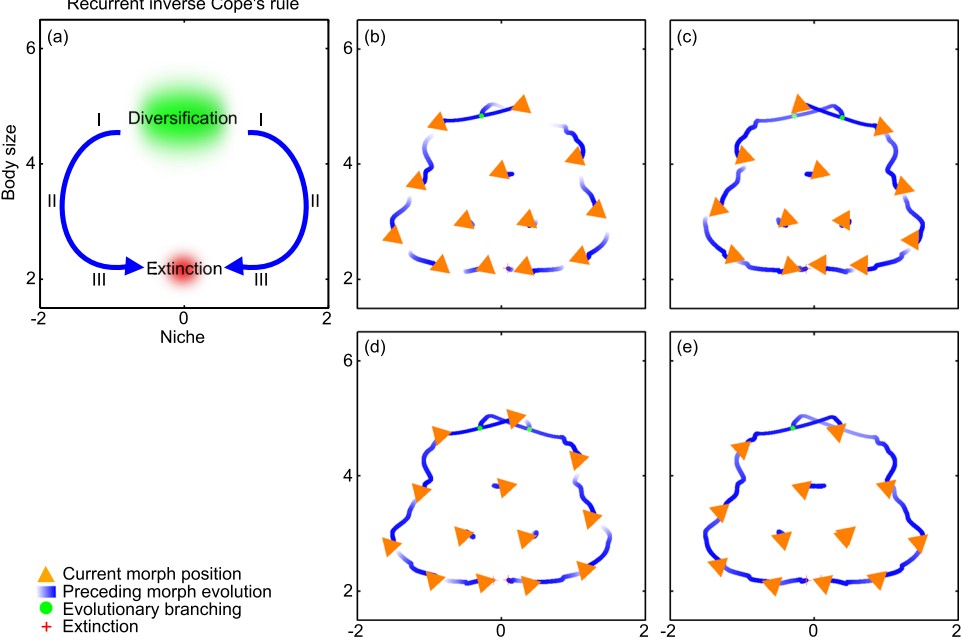

**Fig. 4 Evolution in body size and niche at high interference competition: recurrent inverse Cope's rule. a** Schematic illustration of phylogenetic pattern. **b–e** Trait combinations in the coevolving community at four successive times. Orange arrowheads show species' current trait values and the current directions of the selection pressures governing their evolutionary change. Blue motion trails represent species' evolutionary trajectories, with their thickness being proportional to species' population densities and with the darkest blue corresponding to the current time. Green circles and red crosses, respectively, indicate past trait combinations at which evolutionary diversifications and species extinctions occurred. In this macroevolutionary pattern, evolutionary diversification occurs at large body sizes: all species then evolve toward smaller body sizes, punctuated by the recurrent extinction of the smallest species. For a full dynamical rendering of the shown community coevolution, see Supplementary Movie S4. Parameters as shown in Table 1, with $\alpha = 0.5$ and $\epsilon = 0.005$.

species with the largest body size next diversify, after which evolution brings about a diversification in ecological niche followed by a reduction in body size. Having attained small body sizes, the species reverse their niche evolution and race to occupy the same niche. After this evolutionary race has inevitably caused extinctions, subsequent diversifications of the top predators perpetuate the process. Because the emergent macroevolutionary pattern is the direct opposite of that described by the recurrent

Cope's rule, we refer to it as following the recurrent inverse Cope's rule (Fig. 4a). And since this counterintuitive pattern has not previously been described in the literature, we detail below the processes and mechanisms contributing to its three phases.

Phase I: Species emerge in pairs from the diversification of an ancestor species with large body size. These then evolve their niches in opposite directions to reduce interference competition while their body sizes remain largely unchanged (Fig. 4a, b). The

strong selection for niche separation occurs at the expense of reducing the energy gained from consuming the basal resource ("Methods"). The diverging species thus reach trait combinations at which their energy gained from the basal resource balances their losses from interference competition, predation, and natural mortality (Fig. 4a, b).

Phase II: As species segregate in ecological niche, the reduction in the energy they can derive from the basal resource causes them to evolve toward smaller body sizes. Once these species have reduced their body size enough to consume substantial amounts of the basal resource directly, strong selection emerges for aligning their ecological niche with that of the basal resource. This reverses their separation in the ecological niche, causing them instead to race toward occupying the basal resource's niche.

Phase III: When the species pair with the lowest body size attain the basal resource's niche, they benefit from consuming the latter, but also suffer from increased predation. At the same time, they come under increasing pressure from the next species pair with roughly the same body sizes that evolve toward the basal resource's niche. The resultant increase in mortality through interference competition eventually drives the former species pair to extinction (Fig. 4b–e), leaving the remaining species to follow the same evolutionary dynamics and perpetuating the process.

**Synthesis**. Our investigation suggests that the evolutionary dynamics of body size determining the validity of Cope's rule depend on two key factors: the strength of interference competition and the risk of species extinction. Figure 5 shows how the three qualitatively different macroevolutionary patterns we have described above arise for different combinations of these factors. Cope's rule applies (gray region in Fig. 5) at low-to-medium levels of interference competition when extinction risk is relatively small (within the range shown in Table 1). If the extinction risk is increased above a threshold, the qualitatively different form of Cope's rule we have called the recurrent Cope's rule applies

(light-blue region in Fig. 5). This regime is characterized by perpetually cyclic coevolutionary dynamics in which top predators go extinct and species nearly always evolve toward larger body sizes. When interference competition is high, we observe the surprising reversal of this pattern according to what we have called the recurrent inverse Cope's rule (light-red region in Fig. 5). While this regime is also characterized by perpetually cyclic coevolutionary dynamics, species evolve toward lower body sizes until selection pressures imposed on them by other species in the coevolving community eventually drive them to extinction.

## Discussion

To understand why species' body sizes have been observed to increase[7–9,11,14–18] or decrease[19–21,27] on evolutionary timescales, we have analyzed whether and when Cope's rule is expected to hold in a process-based community-evolution model that accounts for the ecophysiological implications of body size and the role of ecological niches. To our knowledge, this is the first time that an ecologically realistic coevolutionary model has been used to investigate the validity of Cope's rule.

We have found three qualitatively different macroevolutionary patterns of which only two are consistent with Cope's rule. The third pattern, first reported here, implies a perpetual phyletic trend toward lower body size. We predict that this novel pattern, which we have dubbed the recurrent inverse Cope's rule, will arise in ecological settings characterized by strong interference competition among species. Our study is providing new insights into the eco-evolutionary mechanisms driving these three contrasting macroevolutionary patterns by identifying the ecological determinants and elucidating the evolutionary consequences—in terms of both phylogenetic patterns and speciation/extinction processes—of Cope's rule and its inverse. Our synthesis reveals that the macroevolutionary patterns that unfold under the ecologically driven community coevolution of body sizes are primarily

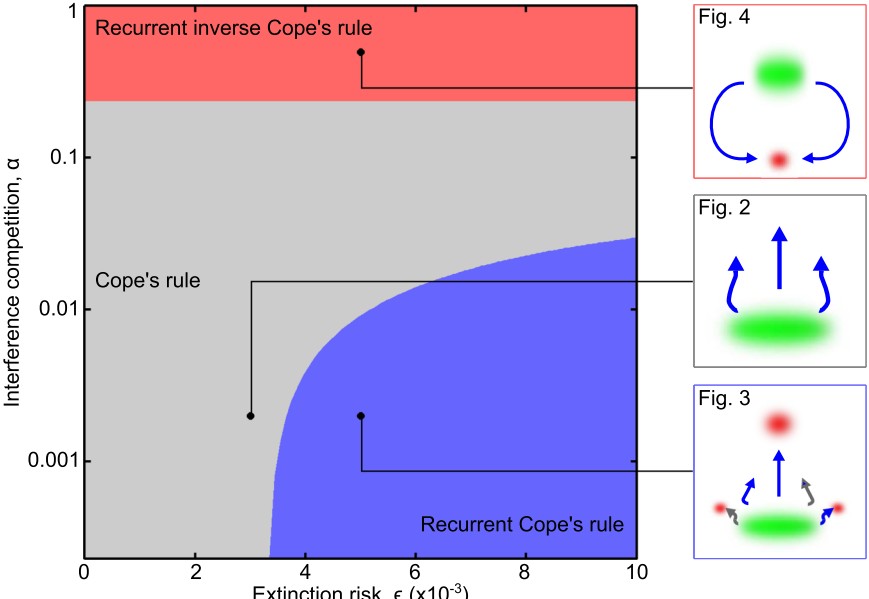

**Fig. 5 Synoptic overview of the three predicted characteristic macroevolutionary patterns and of the different ecological conditions under which they emerge.** Our process-based community-evolution model describing joint adaption and coevolution in the body sizes and ecological niches of a community's species reveals the key roles played by the strength of interference competition and the risk of population extinction in determining qualitatively different patterns of body-size evolution. The boundaries between the three shown regions are determined by running our model for different combinations of $\alpha$ and $\epsilon$, manually classifying the resulting phylogenies according to the three macroevolutionary patterns, and interpolating the results to produce smooth curves. Black circles indicate the parameter combinations illustrated in Figs. 2–4, with the corresponding macroevolutionary patterns summarized in the three small panels on the right-hand side.

determined by just two key ecological factors: strength of interference competition and level of extinction risk.

We have demonstrated that when adaptations are restricted to body sizes, i.e., in the absence of niche differentiation based on other evolving traits, the species comprising coevolving communities are expected to increase in body size until reaching a coevolutionary equilibrium at which selection for larger body sizes ceases. Cope's rule thus applies and is most noticeable early on in such a community's phylogenetic history. For sufficiently high extinction risk, this pattern changes into the recurrent Cope's rule, according to which species perpetually keep evolving toward larger body sizes, as they successively take the place of large top predators that have become extinct. The possibility of perpetual evolution toward larger body size driven by extinctions, has in fact been predicted before, based on qualitative verbal reasoning, by Brown and Mauer[47], who argued that, as a consequence of selection for larger body sizes, "giant forms are produced, but these frequently become extinct, creating new opportunities for another lineage as it in turn evolves organisms of larger size." This conclusion applies independently of the strength of interference competition, which only has the quantitative consequence of increasing the number of species in the coevolving community. Surprisingly, by introducing the possibility for species adaptively to alter their ecological niche, a new, third pattern, which we have dubbed the recurrent inverse Cope's rule becomes possible. We hope that future research will succeed in assessing the extent to which this model-predicted macroevolutionary pattern is consistent with known empirical counterexamples of Cope's rules, e.g., size declines in cryptodiran turtles[20], Alaskan Pleistocene horses[24], island lizards[23], and Mississippian vertebrates after the end-Devonian extinction[22].

For comparing our findings with empirical observations, it is important to appreciate that the niche trait we have analyzed is representing a plethora of different concrete traits describing salient niche differentiation in different ecological systems. Examples abound and include habitat preferences or tolerances with respect to environmental factors such as temperature, precipitation, irradiation, latitude, terrestrial altitude, aquatic depth, water-flow velocity, vertical canopy position, topographical slope, microbiome composition, soil type, disturbance regime, growth season, or geographical range. Building on mounting data covering such factors will likely yield insights into how to develop our model and apply it with enhanced realism to specific ecological systems. As a first step in this direction, one could, for example, define the niche traits of species based on their geographical ranges. Empirical data on current and historical geographical ranges are increasingly available (e.g., refs. [57,58]) and could be used for this purpose. Specifically, one could use the average or typical location of individuals in a species for defining their niche traits. Moreover, as promising extensions of our research, it would be interesting to allow for species-dependent and/or adaptive widths of the modeled ecological niches[59] with the aim of thereby facilitating and strengthening comparisons with empirical observations.

In the current version of our model, intraspecific variation in body sizes and niche traits has not been explicitly incorporated. Considering such intraspecific variation can impact the evolution of traits, as demonstrated by, e.g., ref. [60]. Intraspecific trait variation can be incorporated into our model by adopting developments in adaptive dynamics theory, such as the oligomorphic dynamics proposed in ref. [61], which allows for the description of quantitative genetic dynamics in asexually reproducing populations with multiple morphs. In addition, recent work in ref. [62] provides a framework for studying how trait diversity is apportioned within and between species in both unstructured and spatially structured populations.

The three different macroevolutionary patterns our analyses have revealed have potential to reconcile and refine the debate on the validity of Cope's rule by providing a richer conceptual framework for assessing and understanding phyletic patterns. In moving beyond merely documenting a trend toward larger body sizes by distinguishing between a gradual increase to an evolutionary equilibrium and a perpetual increase driven by extinctions, it may be possible to focus paleontological efforts on taxa and systems that are known to have higher extinction risks and in which selection pressures for higher body sizes may be larger. Thus, our study is opening up possibilities for mapping ecological determinants and process-based mechanisms onto the contrasting patterns in body-size evolution observed in nature. In this way, several testable hypotheses can be formulated based on the findings we have reported here.

First, our study puts forward the innovative hypothesis that the prevalence of Cope's rule or its inverse crucially depends on ecological interactions and the rate at which niche utilization evolves over time. Accordingly, we suggest that for understanding the evolution of body sizes across taxa, due consideration needs to be given to rates of niche evolution and levels of interference competition. Second, applying our findings in reverse, we further hypothesize that an important reason why Cope's rule has not been observed across all investigated lineages[20,21,27] is that the species in the considered ecological systems have simultaneously adapted in terms of ecological niches other than those merely determined by their body size. In other words, researchers of Cope's rule need to recognize the possibility that evidence of such complementary niche evolution has gone unnoticed in the paleontological record. Finally, two more hypotheses can be put forward with regard to the contrasting patterns of species extinction: recurrent extinctions of the largest taxa are a consequence of strong predatory interactions leading to the evolution of very large species that are particularly vulnerable to environmental fluctuations, while recurrent extinctions of the smallest taxa[3,54,56] are a consequence of strong interference competition. We recognize that the paleontological record has much to offer to enable testing the hypotheses proposed by this study. This is delineating a necessary next step forward in understanding the contrasting macroevolutionary patterns observed in the coevolution of body sizes.

## Methods
We determine phylogenies using a process-based community-evolution model that describes changes in two adaptive traits, body size and ecological niche. Body size is a key functional trait with well-documented ecological implication (e.g., ref. [48]), and adaptation of this trait alone enables the emergence of trophically structured communities[44,46] (see also the review in ref. [63]). The similarity in ecological niche plays a fundamental complementary role in scaling species interactions, with interaction strengths naturally being maximized among individuals occupying the same niche. Accounting for this second trait in our model is a critical prerequisite for more complex processes of evolutionary diversification and, therefore, for the emergence of richer and more realistic community structures[64]. Below, we explain how these two traits jointly determine demographic dynamics and how gradual adaptive change over evolutionary time creates complex trophically structured ecological communities, complete with their specific phylogenetic histories.

**Demographic dynamics.** We consider communities comprising $N$ heterotrophic species designated by the indices $i = 1, \ldots, N$ that are interacting among each other and with one basal autotrophic resource designated by the index $i = 0$. The community's

species richness $N$ is changing dynamically, through processes of extinction and speciation, as detailed below. Each species $i$ is characterized by its population density $x_i$ and two adaptive traits describing the average body size $s_i$ and ecological niche $n_i$ of its individuals. Following Brännström et al.[44], we express $s_i$ in nondimensional logarithmic form as $r_i = \ln(s_i/s_0)$, where $s_0$ is the size of the basal autotrophic resource. While population densities and body sizes are necessarily non-negative, niche traits can take non-negative and negative values. We fix the otherwise arbitrary origin of the niche traits by assuming $n_0 = 0$ for the basal autotrophic resource without loss of generality. All model parameters are shown in Table 1 together with their default values.

The demographic dynamics of the $N$ heterotrophic species $i = 1, \dots, N$ and of the one basal autotrophic resource $i = 0$ are described by Lotka–Volterra equations,

$$
\overbrace{\frac{\dot{x}_i}{x_i}}^{\text{Growth}} = -\overbrace{d(r_i)}^{\text{Intrinsic mortality}} + \overbrace{\sum_{j=0}^{N} \beta P(n_i, n_j) p(r_i, r_j) \lambda \exp(r_j - r_i) x_j}^{\text{Gains from predation}}
$$
$$
-\overbrace{\sum_{j=1}^{N} \beta P(n_j, n_i) p(r_j, r_i) x_j}^{\text{Losses from predation}} - \overbrace{\sum_{j=1}^{N} \alpha C(n_i, n_j) c(r_i, r_j) x_j}^{\text{Losses from competition}}
$$

(1a)

and

$$
\overbrace{\frac{\dot{x}_0}{x_0}}^{\text{Growth}} = +\overbrace{g_0}^{\text{Intrinsic growth}} - \overbrace{\sum_{j=1}^{N} \beta p(r_j, 0) P(n_j, 0) x_j}^{\text{Losses from predation}} - \overbrace{x_0/K_0}^{\text{Losses from competition}},
$$

(1b)

where $\dot{x}_i$ denotes the rate at which the population density $x_i$ changes. The terms on the right-hand side of Eq. (1a) are the per-capita rates of, for the heterotrophic species, intrinsic mortality, gains from predation, losses from predation, and losses from interference competition, respectively. Similarly, the terms on the right-hand side of Eq. (1b) are the per-capita rates of, for the basal autotrophic resource, intrinsic growth, losses from predation, and losses from competition, respectively. Gains can be realized through increased fecundity, reduced mortality, or a mixture of both, and, likewise, losses can be realized through reduced fecundity, increased mortality, or a mixture of both.

We consider a species to be extant as long as its population density exceeds the threshold $\epsilon$; conversely, if and when a species' population density falls below this threshold, it is considered extinct and is removed from the community. The parameter $\epsilon$ can thus be interpreted as a measure of extinction risk resulting from sensitivity to demographic and environmental stochasticity.

The rate of intrinsic mortality and the intensities of predation and interference competition depend on the two adaptive traits. To reflect the energetic advantages of a larger body size over a smaller one, the intrinsic mortality rate is assumed to decline allometrically with the body size $s_i$, and thus exponentially with the logarithmic body size $r_i$, according to an exponent $q$, whose value is suggested by Peters[48] to equal ~0.25,

$$
d(r_i) = d_0 (s_i/s_0)^{-q} = d_0 \exp(-q r_i). \tag{2a}
$$

The intensities of predation and interference competition between individuals of two species $i$ and $j$ occupying the same niche, $n_i = n_j$, are determined by the ratio of their body sizes $s_i$, and thus by the difference of their logarithmic body sizes $r_i$. A predator of species $i$ and logarithmic body size $r_i$ forages on a prey of species $j$ and logarithmic body size $r_j$ at an intensity that is assumed to be maximized when their logarithmic body sizes differ by a value $\mu$ that is optimal for predation,

$$
p(r_i, r_j) = \exp\left(-\tfrac{1}{2}(r_i - r_j - \mu)^2/\sigma_P^2 - \tfrac{1}{4}(r_i - r_j)^4/\gamma_P^4\right). \tag{2b}
$$

Similarly the intensity of interference competition between individuals of two species $i$ and $j$ occupying the same niche and having logarithmic body sizes $r_i$ and $r_j$ is assumed to be symmetrical and maximal for individuals of equal body size,

$$
c(r_i, r_j) = \exp\left(-\tfrac{1}{2}(r_i - r_j)^2/\sigma_c^2 - \tfrac{1}{4}(r_i - r_j)^4/\gamma_c^4\right). \tag{2c}
$$

The intensities of predation and interference competition, respectively, between individuals of two species $i$ and $j$ occupying different niches, $n_i \neq n_j$, are reduced by factors described by functions that decline with increasing niche separation,

$$
P(n_i, n_j) = \exp\left(-\tfrac{1}{2}(n_i - n_j)^2/\sigma_P^2 - \tfrac{1}{4}(n_i - n_j)^4/\gamma_P^4\right) \tag{2d}
$$

and

$$
C(n_i, n_j) = \exp\left(-\tfrac{1}{2}(n_i - n_j)^2/\sigma_C^2 - \tfrac{1}{4}(n_i - n_j)^4/\gamma_C^4\right). \tag{2e}
$$

To ensure our results are robust when the functions above deviate from Gaussian shapes, we allow platykurtic functions in Eqs. (2c)–(2e): specifically, the parameters $\gamma_P$, $\gamma_c$, $\gamma_P$, and $\gamma_C$ scale the quartic terms in the exponents above and hence the extent to which those functions are platykurtic, i.e., deviate from Gaussian shapes in the direction of more box-like shapes. Even slight degrees of platykurtosis are known to overcome the historically often overlooked structural instability caused by purely Gaussian functions in models of trait-mediated competition and thereby suffice to enable the ecologically and evolutionarily stable coexistence of phenotypically differentiated discrete species (e.g., refs. [65,66]).

In summary, the combined effects of body size and ecological niche on predation and interference competition are given by the products $p(r_i, r_j) P(n_i, n_j)$ and $c(r_i, r_j) C(n_i, n_j)$, respectively, as shown in Eqs. (1).

**Evolutionary dynamics.** The evolutionary dynamics of the adaptive traits are determined by the corresponding selection pressures (e.g., refs. [44,45]). Writing $F(N; x_0, \dots, x_N; s_0, \dots, s_N; n_0, \dots, n_N)$ for the right-hand side of Eq. (1a), we define the invasion fitness of an initially rare population with trait values $s'$ and $n'$ in a community comprising the autotropic basal resource and $N$ resident heterotrophic species with population densities $x_0, \dots, x_N$ and trait values $s_0, \dots, s_N$ and $n_0, \dots, n_N$ as

$$
f(N; x, s, n; s', n') = \lim_{x' \to 0+} F(N+1; x_0, \dots, x_N, x'; s_0, \dots, s_N, s'; n_0, \dots, n_N, n'),
$$

(3a)

where $x = (x_0, \dots, x_N)$, $s = (s_0, \dots, s_N)$, and $n = (n_0, \dots, n_N)$.

We solve the $N+1$ demographic equations in Eqs. (1) alongside $2N$ evolutionary equations, one for each trait in each species,

$$
\dot{s}_i = \varepsilon_s \left. \frac{\partial f(N; x, s, n; s', n')}{\partial s'} \right|_{s'=s_i, n'=n_i} \tag{3b}
$$

and

$$
\dot{n}_i = \varepsilon_n \left. \frac{\partial f(N; x, s, n; s', n')}{\partial n'} \right|_{s'=s_i, n'=n_i}, \tag{3c}
$$

where $\varepsilon_s$ and $\varepsilon_n$ scale the rates of evolutionary change. We assume $\varepsilon_s$ and $\varepsilon_n$ to be so small that body sizes and ecological niches are evolving slowly relative to the demographics dynamics.

Evolution of the adaptive traits under directional selection proceeds according to Eqs. (3) until a local fitness minimum is encountered in one or more of the heterotrophic species and selection thus turns disruptive. Specifically, we test whether the magnitudes of the selection pressures, i.e., of the derivatives in

Eqs. (3b) and (3c), fall below a prescribed threshold for both adaptive traits. If and when the underlying extremum in a species' invasion-fitness landscape given by Eq. (3a) happens to be a minimum, the species is replaced with two species with trait values shifted a fixed distance toward either side of the fitness minimum along the direction of steepest increase (i.e., highest curvature) of invasion fitness, in a process intended to mimic ecological speciation[67,68].

**Reporting summary**. Further information on research design is available in the Nature Portfolio Reporting Summary linked to this article.

## Data availability
All data supporting the results and conclusion are included within the article. Data shown in the figures were generated through model runs using MATLAB.

## Code availability
The model code has been deposited to the University of Reading's Open Access repository and can be obtained freely from https://doi.org/10.17864/1947.000491.

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

## Acknowledgements

S.R. acknowledges support by the Evolution and Ecology Program at IIASA for two research visits facilitating the development of this study. This work was a part of S.R.'s ongoing research at the University of Reading. U.D. acknowledges financial support by the European Commission, the European Science Foundation, the Austrian Science Fund, the Austrian Ministry of Science and Research, the Vienna Science and Technology Fund, and the Global Bioconvergence Center of Innovation at the Okinawa Institute of Science and Technology Graduate University, OIST, supported by a grant from the Japan Science and Technology Agency, JST, Program for an Open Innovation Platform for Academia-Industry Co-Creation, COI-NEXT. S.R., Å.B., and U.D. acknowledge funding from IIASA and the National Member Organizations that support the institute.

## Author contributions

S.R., Å.B., and U.D. developed the model. S.R. wrote the computer code, performed simulations of the model, generated figures, and prepared a first draft of the paper. S.R., Å.B., and U.D. developed the results and contributed to the writing of the final paper.

## Competing interests

The authors declare no competing interests.
