## [Peer Review File · Communications Biology]

** Please ensure you delete the link to your author homepage in this email if you wish to forward it to your coauthors **

Dear Dr Roy,

Your manuscript entitled "Ecological determinants of Cope's rule and its inverse" has now been seen by 2 referees. You will see from their comments below that while they find your work of considerable interest, some important points are raised. We are interested in the possibility of publishing your study in *Communications Biology*, but would like to consider your response to these concerns in the form of a revised manuscript before we make a final decision on publication.

We therefore invite you to revise and resubmit your manuscript, taking into account the points raised. In particular, some revisions are required to the introduction to address the reviewer 2's comments regarding defining some of the terminology for ease of reading, and also some revisions to the discussion to address reviewer 1's comments about the conclusions drawn.

Please highlight all changes in the manuscript text file.

We are committed to providing a fair and constructive peer-review process. Do not hesitate to contact us if you wish to discuss the revision in more detail or if there are specific requests from the reviewers that you believe are technically impossible or unlikely to yield a meaningful outcome.

At the same time, we ask that you ensure your manuscript complies with our editorial policies. Please see [our revision file checklist](https://www.nature.com/documents/CommsBio-file-checklist-revision.pdf) for guidance on formatting the manuscript and complying with our policies. You will also find guidelines for replying to the referees' comments. You may also wish to review our formatting guidelines for final submissions [here](https://www.nature.com/documents/commsj-life-style-formatting-guide-accept.pdf).

Please use the following link to submit your revised manuscript, point-by-point response to the referees' comments (which should be in a separate document to the cover letter) and any additional files:

<https://mts-commsbio.nature.com/cgi-bin/main.plex?el=A1Cx2DeJ1A2srW5I3A9ftddHSKSq4ZKI7Kc6Hv8OyZNwZ>

When submitting the revised version of your manuscript, please pay close attention to our [Digital Image Integrity Guidelines](https://www.nature.com/commsbio/editorial-policies/image-integrity) and to the following points below:

- that unprocessed scans are clearly labelled and match the gels and western blots presented in figures.

- that control panels for gels and western blots are appropriately described as loading on sample processing controls

- all images in the paper are checked for duplication of panels and for splicing of gel lanes.

We would expect revisions of this nature to take around three months, but appreciate that every situation is unique. We look forward to receiving your revised manuscript when it is ready, and will not enforce a hard deadline on this revision.

Please do not hesitate to contact me if you have any questions or would like to discuss these revisions further. We look forward to seeing the revised manuscript and thank you for the opportunity to review your work.

Best regards,

Katie Davis, PhD
Editorial Board Member
Communications Biology
orcid.org/0000-0002-9235-7853

Referee expertise:

Referee #1: Computational evolutionary biology

Referee #2: Macroevolutionary modelling, diversification, population dynamics

Reviewers' comments:

Reviewer #1 (Remarks to the Author):

I thought that this article took on an interesting angle on what has become a somewhat tired question: Cope's rule. I really like the approach of using community modelling to tackle deep-time dynamics, and I think the authors accomplish it well. I believe that the paper should be published, but I only have a couple of minor comments.

My overarching thought is that, while I appreciate the work, I am not sure that I agree with the authors' conclusions. It is not clear to me that their work has *_really_* provided the level of insight into Cope's rule that they imply, although I feel that the work is valuable and its general framework does itself constitute a substantial contribution. My main issue is that I feel that the authors 1) make many strong assumptions about the nature of speciation and phenotypic evolution-- particularly that these processes are driven predominantly by competitive ecological dynamics. This is, of course, itself a major question in ecology and evolution, and so it leads to this paper feeling a bit like it attempts to solve one somewhat minor problem (Cope's rule) by making very strong assumptions about a fairly foundational open question. This is not inherently a bad thing-- there are of course many excellent workers who would not only accept, but agree with the author's assumption. And it is of course valuable in and of itself to explore how particular dynamics can follow from one set of assumptions about evolutionary and ecological processes.

That being said, my agreeing/disagreeing of course, does not preclude publication. I think the most substantive criticism that I'd have is that the study design is such that it results in some conclusions that feel like truisms based on the design of the model/simulations. For example:

"We thus conclude that when species share a niche and their ecological interactions are determined by their body size, they primarily evolve toward larger body size, as predicted by Cope's rule."

This is of course exactly what one would expect if setting up the scenario in that way. The same goes for the "recurrent" Cope's rule. To my mind, the third pattern, the "recurrent, inverse" Cope's rule, feels like the most counter-intuitive, but it still feels in part like a product of the model. I'm not really sure what I suggest to this end, but it makes me question how much meaningful insight this contribution brings into the biological question at hand.

Nevertheless, I appreciate the authors' approach of predicting emergent large-scale patterns from process-based models. It represents a more satisfying angle than much of the pretty repetitive Cope's rule literature and is something that I feel could benefit the field. As a result, I endorse its publication, and hope that it will influence future work examining the dynamics that come into play as lower-level processes generate emergent patterns. To be honest, I wish that the authors would have submitted this to a more topical journal, like *American Naturalist* or *Evolution*. I believe that would have actually increased the reach and impact of this paper, which I think could be valuable for the field.

Minor notes:

L. 4: Cope's rule posits that evolution gradually increases the body size of animals in lineages.

This is a strange sentence. Aren't all animals in lineages?

L. 145: "how sufficiently high extinction risks qualitative alter the emergent phylogenetic pattern."

wording

Reviewer #2 (Remarks to the Author):

This study uses simulations of species and trait evolution to investigate the processes underlying Cope's rule (the evolutionary tendency for animals to increase in their body size). The authors find that depending on the drivers of species interaction (determined by trait and/or niche overlap) and levels of extinction risks, species may evolve under a Cope's rule, or following different patterns, such as the recurrent evolution of larger body sizes with extinction-driven turnover. The paper provides an interesting theoretical basis to explain empirical evidence (or lack of it) for Cope's rule based on a mechanistic simulation framework.

While I think the overall approach used in this paper and its findings are novel and interesting there are several issues (listed below) that I think should be considered in revising this study.

I understand that this is the manuscript format for the journal, but I personally found reading the Results before the methods quite hard because the underlying machinery is not clear, and it is not possible to understand what mechanisms are driving the patterns shown in the figures from just reading the Results. The methods are themselves quite cryptic and it is not always clear what aspects of the simulation are new to this paper and what aspects are borrowed from previous models and implementations. I think the paper should include a section explaining upfront what the simulations are doing and what the terms used in the Results mean. For instance: what does "interference competition" mean? What does the "extinction risk" parameter represent and what mechanisms drive species to extinction? The figures show diversification events, which presumably represents speciation: how is that modeled? Is speciation splitting an ancestral population in two? (it looks like there are three descendants in Fig. 2). Also, clarifying early on in the paper whether the simulations are stochastic or deterministic would be useful.

One limitation of these simulations is (as far as I can tell) the lack of spatial processes and the assumption that all species can compete if they occupy the same niche. In the real world, species have limited geographic ranges and variable dispersal ability, making it possible for species to avoid competition by dispersing to a new area. This is something that should be discussed in the paper.

Can the parameter values be interpreted in absolute or relative terms? For instance, when referring to a 'high interference competition' (line 171) is this relative to some other parameter? Similarly, the term 'level of extinction risk' sounds like vague, can the author(s) help the reader interpreting this value?

Line 289: While I appreciate the use of simulations to produce process-based expectations, it is difficult to see how empirical research can use this simulation framework to test specific hypotheses. Can the author(s) provide examples? In line 286 it is not clear what the author(s) refer to: what are taxa with known higher extinction risk? Extinction risks are arguably varying significantly through time and I am not sure specific taxa can be identified as inherently exposed to high extinction risk.

Line 60 and elsewhere: I am not a fan of these type of statements of novelty, which I find really hard

to demonstrate. Does anybody have a full overview of all papers published to make such claim? I think it is also unnecessary as this paper's novelty is apparent without these statements.

Figure 5: are the boundaries shown by the different colors determined from the equations or based on simulations? Or are they showing rule-of-thumbs thresholds? All options are fine but this should be clarified in the caption.

Data availability: I think 'data available upon request' is an outdated approach to ensure data availability and one that does not guarantee transparent access to them (authors may change email address or job or become unavailable). Codes and data should be provided either as supplementary data attached to the paper or in a permanent open-access repository (e.g. hosted by Dryad or Zenodo) with a DOI cited in the paper.

Response to reviewers

Reviewer #1

I thought that this article took on an interesting angle on what has become a somewhat tired question: Cope's rule. I really like the approach of using community modelling to tackle deep-time dynamics, and I think the authors accomplish it well. I believe that the paper should be published, but I only have a couple of minor comments.

→ We thank the reviewer for their time and valuable comments. We have tried our best to address them in the revised manuscript. Below, we describe in detail the changes we have made.

My overarching thought is that, while I appreciate the work, I am not sure that I agree with the authors' conclusions. It is not clear to me that their work has _really_ provided the level of insight into Cope's rule that they imply, although I feel that the work is valuable and its general framework does itself constitute a substantial contribution. My main issue is that I feel that the authors 1) make many strong assumptions about the nature of speciation and phenotypic evolution-- particularly that these processes are driven predominantly by competitive ecological dynamics. This is, of course, itself a major question in ecology and evolution, and so it leads to this paper feeling a bit like it attempts to solve one somewhat minor problem (Cope's rule) by making very strong assumptions about a fairly foundational open question. This is not inherently a bad thing-- there are of course many excellent workers who would not only accept, but agree with the author's assumption. And it is of course valuable in and of itself to explore how particular dynamics can follow from one set of assumptions about evolutionary and ecological processes.

That being said, my agreeing/disagreeing of course, does not preclude publication. I think the most substantiative criticism that I'd have is that the study design is such that it results in some conclusions that feel like truisms based on the design of the model/simulations. For example:

"We thus conclude that when species share a niche and their ecological interactions are determined by their body size, they primarily evolve toward larger body size, as predicted by Cope's rule."

This is of course exactly what one would expect if setting up the scenario in that way. The same goes for the "recurrent" Cope's rule. To my mind, the third pattern, the "recurrent, inverse" Cope's rule, feels like the most counter-intuitive, but it still feels in part like a product of the model. I'm not really sure what I suggest to this end, but it makes me question how much meaningful insight this contribution brings into the biological question at hand. Nevertheless, I appreciate the authors' approach of predicting emergent large-scale patterns from process-based models. It represents a more satisfying angle than much of the pretty repetitive Cope's rule literature and is something that I feel could benefit the field. As a result, I endorse its publication, and hope that it will influence future work examining the dynamics that come into play as lower-level processes generate emergent patterns.

→ We are grateful for the reviewer's insightful remarks. We largely agree with these remarks and have therefore tried to better emphasize and motivate our model assumptions. As the reviewer states, we indeed attempt to provide a fresh angle on the age-old Cope's rule, by looking at it from the perspective of a process-based model.

In the revised manuscript, we now include a new section "Model overview" right before the Results section in which we provide a model overview and carefully describe and motivate

our fundamental assumptions. Specifically, we emphasize the generality and commonness of our ecological assumptions and highlight that our model incorporates both trophic and non-trophic interactions encompassing all three fundamental types of ecological competition: exploitative competition, interference competition, and apparent competition.

We have also reworded the sentence quoted by the reviewer to bring out its meaning more clearly.

While our approach, as that of any process-based model, requires specific assumptions and thus comes with associated limitations, we hope – like the reviewer also seems to do – that it is valuable to document the micro- and macroevolutionary consequences of general and common ecological assumptions and that our work hence has potential to stimulate and inspire new research in ways that are complementary to what purely empirical studies could previously do.

To be honest, I wish that the authors would have submitted this to a more topical journal, like American Naturalist or Evolution. I believe that would have actually increased the reach and impact of this paper, which I think could be valuable for the field.

→ We appreciate this suggestion and will consider American Naturalist and Evolution – both excellent journals – for our future work. In this case, we believe Communications Biology with its open access will be helpful to reach a wider audience. Also, having published in both American Naturalist and Evolution in the past, we are interested in trying a new potential outlet for our work.

Minor notes:

L. 4: Cope's rule posits that evolution gradually increases the body size of animals in lineages.

This is a strange sentence. Aren't all animals in lineages?

*L. 145: "how sufficiently high extinction risks qualitative alter the emergent phylogenetic pattern."
wording*

→ We thank the reviewer for these two notes and have revised the corresponding text for improved clarity.

Reviewer #2

This study uses simulations of species and trait evolution to investigate the processes underlying Cope's rule (the evolutionary tendency for animals to increase in their body size). The authors find that depending on the drivers of species interaction (determined by trait and/or niche overlap) and levels of extinction risks, species may evolve under a Cope's rule, or following different patterns, such as the recurrent evolution of larger body sizes with extinction-driven turnover. The paper provides an interesting theoretical basis to explain empirical evidence (or lack of it) for Cope's rule based on a mechanistic simulation framework.

While I think the overall approach used in this paper and its findings are novel and interesting there are several issues (listed below) that I think should be considered in revising this study.

→ We thank the reviewer for his/her time and valuable comments. We have tried our best to address them in the revised manuscript. Below, we describe in detail the changes we have made.

I understand that this is the manuscript format for the journal, but I personally found reading the Results before the methods quite hard because the underlying machinery is not clear, and it is not possible to understand what mechanisms are driving the patterns shown in the figures from just reading the Results. The methods are themselves quite cryptic and it is not always clear what aspects of the simulation are new to this paper and what aspects are borrowed from previous models and implementations. I think the paper should include a section explaining upfront what the simulations are doing and what the terms used in the Results mean. For instance: what does “interference competition” mean? What does the “extinction risk” parameter represent and what mechanisms drive species to extinction? The figures show diversification events, which presumably represents speciation: how is that modeled? Is speciation splitting an ancestral population in two? (it looks like there are three descendants in Fig. 2). Also, clarifying early on in the paper whether the simulations are stochastic or deterministic would be useful.

→ As the reviewer notes, we have followed the manuscript structure supported by the journal, and therefore the detailed methodological description, including model equations, is placed at the end. Implementing the reviewer’s excellent suggestion, we now include a new section “Model overview” right before the Results section, in which we describe our model assumptions in words, provide a better historical context, present our terminology, and discuss the specific points raised by the reviewer. This includes descriptions of interaction types, extinction risk, evolutionary adaptation under directional selection, and evolutionary diversification under disruptive selection.

Below, we directly respond to the reviewer’s last two comments:

First, speciation happens in our model when an ancestral population splits into two.

Naturally, this can lead to three populations when a single population initially splits into two populations, following which only one of those two populations then further splits into two populations. This is what can be seen in Fig. 2: diversification first happens in the body-size trait, after which the species with lower body size diversifies in the niche trait.

Second, our simulations are deterministic, as we now clearly state in the new section “Model overview”.

One limitation of these simulations is (as far as I can tell) the lack of spatial processes and the assumption that all species can compete if they occupy the same niche. In the real world, species have limited geographic ranges and variable dispersal ability, making it possible for species to avoid competition by dispersing to a new area. This is something that should be discussed in the paper.

→ We fully agree with this comment and have updated our manuscript accordingly. In the newly inserted fourth paragraph of the Discussion section, we now describe in much greater detail that, while our model does not explicitly include spatial processes, the niche trait we analyze can capture competition for space or habitats by representing habitat preferences or tolerances with respect to environmental factors as diverse as temperature, precipitation, irradiation, latitude, terrestrial altitude, aquatic depth, water-flow velocity, vertical canopy position, topographical slope, microbiome composition, soil type, disturbance regime, growth season, or geographical range.

Can the parameter values be interpreted in absolute or relative terms? For instance, when referring to a ‘high interference competition’ (line 171) is this relative to some other parameter? Similarly, the term ‘level of extinction risk’ sounds like vague, can the author(s) help the reader interpreting this value?

→ The parameter values are given in Table 1 and are interpreted in absolute terms. We now clarify this in the text referring to the table. Also, the meaning of extinction risk is now clarified in the new “Model overview” section.

Line 289: While I appreciate the use of simulations to produce process-based expectations, it is difficult to see how empirical research can use this simulation framework to test specific hypotheses. Can the author(s) provide examples? In line 286 it is not clear what the author(s) refer to: what are taxa with known higher extinction risk? Extinction risks are arguably varying significantly through time and I am not sure specific taxa can be identified as inherently exposed to high extinction risk.

→ Based on our current work, the most obvious lead for using the reported process-based expectations would be to search for the predicted phylogenetic patterns in the paleontological record. A second step would be to seek for correlations between these patterns and the extinction risk or strength of interference competition, i.e., trying to test the predictions provided in Fig. 5 of our manuscript.

In addition, future work that incorporates body-size dependent extinction risks may be able to provide a wider range of predictions, some of which could again be tested empirically. We now discuss these possibilities in a newly inserted fifth paragraph of the Discussion section.

Line 60 and elsewhere: I am not a fan of these type of statements of novelty, which I find really hard to demonstrate. Does anybody have a full overview of all papers published to make such claim? I think it is also unnecessary as this paper’s novelty is apparent without these statements.

→ We agree with the reviewer that it is not possible to have a full overview of all published papers. Consequently, we have revised the sentence to reflect that the statement made is only based on the best of our knowledge.

Figure 5: are the boundaries shown by the different colors determined from the equations or based on simulations? Or are they showing rule-of-thumbs thresholds? All options are fine but this should be clarified in the caption.

→ The boundaries are based on numerous full model runs. As each model run is computationally expensive, we carry them out on a finite grid of parameter combinations and then use interpolation to determine the boundaries between the regions. This is now explicitly stated in the newly inserted third sentence of the figure caption.

Data availability: I think ‘data available upon request’ is an outdated approach to ensure data availability and one that does not guarantee transparent access to them (authors may change email address or job or become unavailable). Codes and data should be provided either as supplementary data attached to the paper or in a permanent open-access repository (e.g. hosted by Dryad or Zenodo) with a DOI cited in the paper.

➔ We fully agree with this. Accordingly, our source code will be published in the University of Reading's open-access repository (<https://researchdata.reading.ac.uk>) with a permanent DOI that will be added in the final version. This is now clearly stated in our revised statement on data availability.

REVIEWERS' COMMENTS:

Reviewer #2 (Remarks to the Author):

I thank the authors for the careful revision of their paper which now reads more clearly. I maintain that this is a very interesting study and I appreciate the importance of using mechanistic simulations to generate macroevolutionary expectations.

I only have a few additional comments.

I would remove the statement of novelty from the Abstract ("Our results provide the first theoretical foundation") as unnecessary and impossible to demonstrate. I think simply removing "first" will do.

I was wondering if intraspecific variance (in phenotypes and niche) is or can be accounted for in the model. Previous simulation work (e.g. doi.org/10.1093/sysbio/syaa055) shows that intraspecific trait variance has an impact on the evolution of traits and their inheritance at speciation and I think it would be worth commenting on this in the manuscript.

In the Discussion, the paragraph about extinction risk in modern species (l. 373-380) can be misleading, at least as currently presented. We should not mistake macroevolutionary expectations with modern patterns that are for the most part shaped by an extremely recent event: humans. Extinction risk in modern species is not determined by natural competition or predation but by habitat loss, over-exploitation, and to a lesser extent climate change. These are not macroevolutionary processes and occur at very different time scales and I think this difference should be clarified in the text.

Response to reviewers

Reviewer #2 (Remarks to the Author):

I thank the authors for the careful revision of their paper which now reads more clearly. I maintain that this is a very interesting study and I appreciate the importance of using mechanistic simulations to generate macroevolutionary expectations. I only have a few additional comments.

→ Thanks for your time and comments. We have considered your final comments carefully as updated the manuscript to the best of our ability.

I would remove the statement of novelty from the Abstract (“Our results provide the first theoretical foundation”) as unnecessary and impossible to demonstrate. I think simply removing “first” will do.

→ Thanks for your view on the novelty. As the authors, it seems important to us to help our readers understand what we believe to be the key original contribution of our article. So, we have decided to amend the phrase, and have considered prefacing our claim of originality with the words “To our knowledge,”.

I was wondering if intraspecific variance (in phenotypes and niche) is or can be accounted for in the model. Previous simulation work (e.g. doi.org/10.1093/sysbio/syaa055) shows that intraspecific trait variance has an impact on the evolution of traits and their inheritance at speciation and I think it would be worth commenting on this in the manuscript.

→ Thanks for pointing out this aspect. We have now added a paragraph in the discussion to mention this point, to provide our readers with constructive suggestions on how intraspecific trait variation can be incorporated in our model, with appropriate references (pp-19 on the track changed version).

In the Discussion, the paragraph about extinction risk in modern species (l. 373-380) can be misleading, at least as currently presented. We should not mistake macroevolutionary expectations with modern patterns that are for the most part shaped by an extremely recent event: humans. Extinction risk in modern species is not determined by natural competition or predation but by habitat loss, over-exploitation, and to a lesser extent climate change. These are not macroevolutionary processes and occur at very different time scales and I think this difference should be clarified in the text.

→ Thanks for your comment. We have agreed that there seems to be a disconnect between the long timescale required for the macroevolutionary dynamics of our model to play out and the short timescale under which recent anthropogenic impacts have shaped and altered species' extinction risks. We therefore decided to leave out this paragraph entirely in revised version.